# The Effects of Pavement Types on Soil Bacterial Communities across Different Depths

**DOI:** 10.3390/ijerph16101805

**Published:** 2019-05-21

**Authors:** Weiwei Yu, Yinhong Hu, Bowen Cui, Yuanyuan Chen, Xiaoke Wang

**Affiliations:** 1State Key Laboratory of Urban and Regional Ecology, Research Center for Eco-Environmental Sciences, Chinese Academy of Sciences, Beijing 100085, China; 18514088911@163.com (W.Y.); huyinhong14@mails.ucas.edu.cn (Y.H.); bwcui_st@rcees.ac.cn (B.C.); 2College of Resources and Environment, University of Chinese Academy of Sciences, Beijing 100049, China; 3Wuhan Botanical Garden, Chinese Academy of Science, Wuhan 430074, China; chenyuanyuan0822@163.com

**Keywords:** impervious pavement, pervious pavement, bacterial community, soil depth, 16S rRNA gene sequencing

## Abstract

Pavements have remarkable effects on topsoil micro-organisms, but it remains unclear how subsoil microbial communities respond to pavements. In this study, ash trees (*Fraxinus Chinensis*) were planted on pervious pavement (PP), impervious pavement (IPP), and non-pavement (NP) plots. After five years, we determined the soil bacterial community composition and diversity by high-throughput sequencing of the bacterial 16S rRNA gene. The results of our field experiment reveal that the presence of pavement changed soil bacterial community composition and decreased the Shannon index, but had no impact on the Chao 1 at the 0–20 cm layer. However, we achieved the opposite result at a depth of 20–80 cm. Furthermore, there was a significant difference in bacterial community composition using the Shannon index and the Chao 1 at the 80–100 cm layer. Soil total carbon (TC), total nitrogen (TN), available phosphorus (AP), NO_3_^−^-N, and available potassium (AK) were the main factors that influenced soil bacterial composition and diversity across different pavements. Soil bacterial composition and diversity had no notable difference between PP and IPPs at different soil layers. Our results strongly indicate that pavements have a greater impact on topsoil bacterial communities than do subsoils, and PPs did not provide a better habitat for micro-organisms when compared to IPPs in the short term.

## 1. Introduction

Impervious pavement (IPP) covered with artificial materials, such as concrete, asphalt, and bricks, is essential to shelter, transportation, and commerce in urban areas [1]. However, they have adverse effects on urban soils, such as compacting the soil [2], depleting carbon and nitrogen storage, and blocking soil-air gas exchange [3]. In addition, previous studies have shown that pavement altered bacterial community composition and decreased bacterial diversity at the topsoil layer [4,5]. However, the effects of pavement on bacterial communities with soil depth remain largely unknown.

Soil micro-organisms are critical for plant growth [6,7] and biogeochemical cycles [8], and there are significant changes in microbial communities with soil depth [9,10]. Soil microbial biomass and enzyme activities decreased with depth in soil covered by vegetation [11] but increased along the depth gradient under IPP [5]. Moreover, the microbial community structure also shifts with depth, and microbial communities in deep soils were found to be relatively similar in several ecosystems [12,13]. Additionally, street trees in the urban landscape provide significant benefits, such as reducing stormwater runoff, improving air quality, and ameliorating the heat-island effect in urban areas [14,15]. Their roots, in general, are sealed by impervious surfaces that affect access to water, air, and nutrients, thus leading them to uptake water and nutrients from deep soil layers. Deep soils may contain microbial communities that are specialized for their environment, are fundamentally distinct from the surface communities [16,17], and form mutualistic relationships with plants [18]. Understanding the effects of pavements on soil microbial communities across different depths can provide a reference for plant growth in urban areas.

It is well-known that using pervious pavements (PPs) as an alternative to IPPs offers one solution to mitigate urban runoff and water pollution [19,20]. Permeable pavements are beneficial to microbial biomass and function diversity compared to impervious surfaces. Some research has suggested that the degree of soil sealing does not significantly affect soil microbial biomass and activity [21], and that PPs do not improve the root zone of mature trees [22]. Thus, it remains unclear whether PPs can provide a better habitat to microbial communities, as well as plants, compared to IPPs.

The objective of this study was to measure the effects of pavements on bacterial communities and to determine whether PPs have fewer negative effects on bacterial communities. We planted ash trees (*Fraxinus Chinensis*), which is a popular urban tree species, on paved (IPP and PP) and non-pavement plots and quantified soil bacterial relative abundance, diversity, and community composition by high-throughput sequencing of the 16S rRNA gene five years later. This experiment addresses the following questions. First, how do bacterial communities respond to pavements across different depths (as deep as 100 cm below the soil surface)? Second, can PPs improve soil microbial habitats?

## 2. Materials and Methods

### 2.1. Experimental Site Description and Design

The study site was in the Zhangtou village, Changping District, Beijing, China (40°12′N, 116°08′E). The mean annual temperature is 12.1 °C, average annual rainfall is 542 mm, and most of the precipitation occurs from June to September. The study land was cultivated for wheat and maize production for many years before the experiment. The soil texture at the test site was sandy loam.

We used a factorial split-plot experimental design to divide the study area into three equal zones for three pavement types: (1) impervious bricks pavement (i.e., IPP), (2) pervious bricks pavement (i.e., PP), and (3) non-pavement (i.e., the control and NP). In each zone, we used three blocks as replicates for planting ash trees (*Fraxinus Chinensis*) with a density of 2.0 m × 2.0 m. The land zones for treatments of PP and IPPs were paved on bricks with a size of 20 cm × 10 cm × 6 cm (length × width × height). The Beijing Yataiyuhong Technology Development Co., Ltd. produced the bricks from a mixture of clay, sand, and coal ash. The pervious capability of pervious bricks is more than 0.4 mm s^−1^. One-year-old ash tree (*Fraxinus Chinensis*) seedlings, which have a similar height (118.5 ± 2.4 cm) and diameter (14.8 ± 1.0 mm), were randomly planted in April 2012.

### 2.2. Soil Sampling

Soil samples were collected in October 2017 to determine the effects of pavement on soil microbial diversity and community composition. At each block, three replicate plots were selected. Five soil cores that were 1 meter apart from the ash tree were collected from each subplot (from a total of nine subplots) from five depth increments: 0–20, 20–40, 40–60, 60–80, and 80–100 cm. The five samples from the same subplot were combined to form a composite sample for each depth. Each sample was placed in a sterile plastic bag, sealed, and transported to the laboratory on ice. After removing the litter layer, roots, and stones, all samples (45 samples) were passed through a 2 mm sieve and then separated into two parts. One subsample was air-dried for analysis of physicochemical properties, and one was stored at −80 °C for DNA extraction.

### 2.3. Determination of Soil Properties

Soil pH was measured using a pH meter (FE20-FiveEasyTM pH, Mettler Toledo, Weilheim, Germany) with a 1:5 (wt/vol) ratio of soil to water following shaking for 30 min. Soil total carbon (TC) and total nitrogen (TN) were determined using the Dumas method by an Element Analyzer (Vario EL III, Elementar, Hanau, Germany). Soil organic carbon (SOC) content was determined by the dry combustion method with an Element Analyzer (Vario EL III, Elementar, Hanau, Germany), using soil pretreated with HCl, and the soil organic matter content was 1.724 × SOC. Nitrate and ammonium were extracted with 2 mol l^−1^ KCl and quantified using a Continuous Flow Analyzer (SAN++, Skalar, and Holand) [23]. Available phosphorus (AP) was extracted with 0.025 mol l^−1^ HCl + 0.03 mol l^−1^ NH_4_F and measured by a visible spectrophotometer [24]. Available potassium (AK) was extracted with NH_4_OAc and determined using ICP-OES [23].

### 2.4. DNA Extraction and High-Throughput Sequencing

Soil DNA was extracted using a commercial kit (FastDNA® SPIN kit for soil, MP Biomedicals, Santa Ana, CA, USA) according to the manufacturer’s instructions, and samples were shipped to Personalbio (Shanghai Personal Biotechnology, Co., Ltd., Shanghai, China) for high-throughput sequencing using the Illlumina MiSeq platform. The V3-V4 region of the bacterial 16S rRNA gene was PCR-amplified using the forward primer 338F (5′-ACTCCTACGGGAGGCAGCA-3′) and the reverse primer 806R (5′- GGACTACHVGGGTWTCTAAT-3′). PCR amplicons were purified using a PCR cleanup kit (Axygen) and quantified using a PicoGreen dsDNA Assay Kit (Invitrogen). Amplicons were then were pooled in an equimolar concentration and sequenced on separate runs for MiSeq using a 2 × 300 bp paired-end protocol. Sequence data are available in the NCBI Sequence Read Archive under accession number from SRR8992492 to SRR8992536.

### 2.5. Sequence Analysis

Raw data were quality-filtered using the Quantitative Insights into Microbial Ecology (QIIME v1.8.0) [25]. Sequences <150 bp and reads containing ambiguous bases or any unresolved nucleotides were removed. Chimeras were identified and removed using USEARCH [26]. Operational taxonomic units (OTUs) were assigned using UCLUST method at similarities of 97%. Any OTU representing less than 0.001% of the total filtered sequences was removed [27]. Indices of community diversity (Chao 1 and the Shannon index) were calculated in MOTHUR (http://www.mothur.org/).

### 2.6. Data Analysis

One-way ANOVAs and two-way ANOVAs were used to analyze depth and pavements effects on soil physio-chemical and microbial properties (LSD; *P* = 0.05). Linear models were conducted to verify relationships between bacterial diversity and soil properties. We determined bacterial community similarity among pavements and soil depth using ANOSIM; non-metric multi-dimensional scaling (NMDS) was used to visualize clusters. Canonical correspondence analysis (CCA) was performed to show a visual relationship between environmental factors and bacterial distributions. The resulting clustering trees were paired with a heatmap of abundance data, created with ‘heatmap.2’ from the ‘gplots’ package. The above statistical analyses were performed using the vegan package of R v.3.1.1 [28] and SPSS software v.18.0 (SPSS Inc., Chicago, IL, USA).

## 3. Results

### 3.1. Bacterial Community Composition

A total of 634,738 quality sequences were obtained across 45 soil samples, and were clustered into 11,597 OTUs after trimming and filtration. The RDP Classifier assigned the sequences to 42 phyla, 193 orders, 329 families, and 536 genera.

The bacterial composition was dominated (5%) by *Actinobacteria*, *Chloroflexi*, *Proteobacteria*, *Acidobacteria*, *Nitrospirae*, and *Gemmatimonadetes*; together this accounted for 85% to 95% of the total sequence data (Figure 1). The pavement altered the relative abundance of dominant phyla (ANOVA, *p* < 0.05), except for *Acidobacteria* at a depth of 0–20 cm. In contrast, there were no significant differences in the relative abundance of the five dominant phyla in the 20–40 cm layer. Only *Acidobacteria* (ANOVA, *p* = 0.018) and *Proteobacteria* (ANOVA, *p* = 0.002) showed significant changes in the 40–60 cm and 60–80 cm layer, respectively. There were significant differences in *Chloroflexi* (ANOVA, *p* = 0.005), *Proteobacteria* (ANOVA, *p* = 0.013), and *Gemmatimonadetes* (ANOVA, *p* = 0.02) at a depth of 80–100 cm.

Differences in the bacterial composition were analyzed using NMDS, based on the Braye–Curtis distance at the OTU level. Microbial communities from IPP were distinctively different from those in PP and NP soils in the 0–20 cm and 80–100 cm layer, as shown in the NMDS plot (Figure 2). The ANOSIM analysis further revealed that only the 0–20 cm layer (R = 0.778, *p* = 0.001) and 80–100 cm (R = 0.407, *p* = 0.009) layer underwent a significant shift in bacterial composition across different pavement types. Soil bacterial composition did not show any significant differences at a depth of 20–40 cm (R = 0.036, *p* = 0.332), 40–60 cm (R = 0.037, *p* = 0.294), and 60–80 cm (R = 0.021, *p* = 0.393). Moreover, no significant differences were found between IPP and PP at each depth in terms of bacterial composition. The ANOSIM analysis also showed that bacterial composition changed with soil depth under IPP (R = 0.6267, *p* = 0.001), PP (R = 0.4370, *p* = 0.012), and NP (R = 0.3452, *p* = 0.032).

A heatmap (with hierarchal clustering), based on the relative abundances of the top 50 genera, was generated to visualize the bacterial community composition better at the genus level (Figure 3). The distinctions were among different pavements, including very high relative abundances of *Pseudomonas* in the IPP and NP sites, and high relative abundance of *Nitrospira* and *Thermoflexus* in the IPP and NP sites. Also worthy of note is the high relative abundance of *Gaiella*, *Solirubrobacter*, and *Bacillus* in the three pavement samples.

### 3.2. Effects of Land Pavement on Soil Bacterial Alpha Diversity

The Chao 1 and the Shannon index were used to compare the levels of bacterial diversity. The Chao 1 and the Shannon index were the highest in the NP plots, and the lowest in the IPP plots at each soil layer. The bacterial richness is represented by the Chao 1 index (Table 1). The bacterial richness did not differ significantly among pavements at the 0–20 cm layer, whereas it was significantly lower in the IPP plots when compared to the NP plots at the other soil layers. When expressed as the Shannon index, diversity was significantly different between IPP sites and NP sites at the 0–20 cm and 80–100 cm layers, but there were no marked differences at the other soil layers. There were also no significant differences in the Chao 1 and the Shannon index except between IPP and PP at an 80–100 cm depth. Stepwise regression demonstrated that bacterial richness correlated with NO_3_^−^-N, AK, and AP. AP, SOC, and AK influenced soil bacterial diversity (Table 2).

### 3.3. Variables Correlated with Bacterial Abundance and Composition

The CCA analysis and Mantel test demonstrated the correlations between the soil bacterial community and the soil properties (Figure 4, Table 3). There was a significant difference in soil properties across different pavements (Appendix A). In the surface soils, distinctions in the bacterial community structure correlated with TC and TN, whereas the observed soil properties were not associated with bacterial composition at the 20–40 cm layer. Soil bacterial structure was affected by NH_4_^+^-N and AP at the 40–60 cm layer and influenced by NO_3_^-^-N and AP at the 60–80 cm layer. At the 80–100 cm soil layer, bacterial community structure had a close relationship with NH_4_^+^-N and NO_3_^−^N. 

The Pearson correlation coefficient evaluated the relationship between edaphic properties and relative phyla abundance (Appendix A). The results showed that the variation of the relative abundances of bacterial phyla had close correlations with soil properties. The phylum *Actinobacteria*, which was one of the most abundant phyla, was positively and significantly correlated with NH_4_^+^-N, TC, and AP concentrations (*p* < 0.05). The phylum *Chloroflexi* had a significant positive relationship with NO_3_^−^-N (*p* < 0.01), and a negative relationship with pH and NH_4_^+^-N (*p* < 0.05). The phylum *Proteobacteria* was influenced by pH, AP, NO_3_^−^-N, and TC (*p* < 0.05).

## 4. Discussion

Our study provides experimental evidence that pavement is a driver of change in topsoil microbial communities, which is coincident with previous observational studies, suggesting that impervious surfaces affect soil microbial community composition, diversity, and biomass [29,30]. However, our study provides the evidence that the effects of pavement on subsoil microbial community are minor in the short-term experiment (i.e., five years of experimentally imposed pavement).

Corroborating the findings of a previous study [4], we showed that pavement disturbance changed the topsoil bacterial community composition. However, the pavement did not affect the soil bacterial community composition at the 20–80 cm layer, which may be because bricks had contact with topsoil directly, and thus had a greater impact on topsoil than subsoil in the short term. Moreover, plants affect the soil microbial community through belowground carbon allocation and nutrient movements [31,32]. Except for NO_3_^−^-N, there was no marked difference in soil properties between pavement and non-pavement at a depth of 20–80 cm in the present study. Correspondingly, plant roots may have a positive influence on microbial community composition by rhizosphere effects at the 20–80 cm layer [33]. It is interesting to note that soil bacterial community composition differed significantly between pavement and non-pavement at the 80–100 cm layer. Owing to the 80–100 cm layer was far from bricks, we could not be sure whether pavement induced these differences, and more experiments are necessary to explore the effects of pavement on deep soils.

Soil microbial diversity plays a critical role in maintaining ecosystem services, including nutrient cycling and litter decomposition [34,35]. Plant roots exert a large impact on soil micro-organisms and their habitat, and microflora had positive feedback plant growth [36,37]. Although previous research shows that both topsoil bacterial diversity and microbial functional diversity decreased beneath impervious surfaces without plants [4,30], we expected pavement near trees to have fewer negative effects on bacterial diversity. Pavement resulted in decreases in topsoil bacteria on the Shannon index rather than the Chao 1 in the present study. In contrast, the pavement had no influence on soil bacteria using the Shannon index, but reduced the bacterial Chao 1 at the 20–80 cm layer. This result demonstrates that bacterial species’ richness and diversity indices respond to pavements in different ways.

The bacterial community structure and diversity changed with artificial disturbance and was affected by abiotic factors [38,39]. Soil carbon and nitrogen were depleted under impervious surfaces [3], which had marked impacts on bacterial communities. Furthermore, microbial activities changed with soil carbon and nitrogen [35,40]. Although the content of carbon and nitrogen under pavement did not reduce sharply during the experimental period, TC and TN were the main factors that changed bacterial community composition at the 0–20 cm layer in the present study. Additionally, TC had a close relationship with the relative abundance of six dominant phyla, including *Actinobacteria, Proteobacteria, Nitrospirae, GAL15, Firmicutes*, and *Planctomycetes*, and TN affected the relative abundance of two dominant phyla. Moreover, ecosystem phosphorus and potassium supply strongly influence soil microbial community structures and nutrient cycling processes [41,42]. Soil bacterial diversity had a close relationship with AP and AK across different pavements in the present study.

PP, used instead of IPP, could benefit urban trees because the permeability to air and water directly affects the soil environment [22]. The effects of PPs on plants, soils, and micro-organisms differ depending on pavement porosity, profile design, and duration of pavements. Hu et al. [4] suggested that permeable pavements (bricks with a hole in the middle) are beneficial to the overall diversity of bacterial communities compared to IPP (concrete) after 10 years. However, Piotrowska-Długosz et al. [21] concluded that microbial biomass experienced no significant difference between impervious and PPs. Furthermore, Viswanathan et al. [22] found that pavements increase soil CO_2_ concentrations and reduce root production, but pervious concrete did not convey any measurable root growth benefits over impervious concrete. PPs changed bacterial richness (Chao 1) rather than bacterial community composition and diversity (Shannon index) in our study. These results demonstrated that PPs had no remarkable effects on bacterial communities when compared to IPPs in the short term.

## 5. Conclusions

In conclusion, our field experiment found that pavements changed topsoil bacterial community compositions and reduced the Shannon index at depths of 0–20 cm and 80–100 cm, but had no significant effects on them at the 20–80 cm layer. In contrast, pavements had reverse effects on the Chao 1 through the soil profile. Furthermore, PP did not alleviate the negative effects of pavements on bacterial communities. Soil nutrients, including TC, TN, SOC, NO_3_^−^-N, AP, and AK, were the key factors that influenced bacterial communities. This study examined soil bacterial communities under ash trees planted for a short term (five years), but plant species and age also are key factors that affect microbial community activity. Future studies should investigate how microbial communities respond to pavements over a longer period. These findings will be highly valuable in guiding the management of soils and plants in urban areas.

## Figures and Tables

**Figure 1 ijerph-16-01805-f001:**
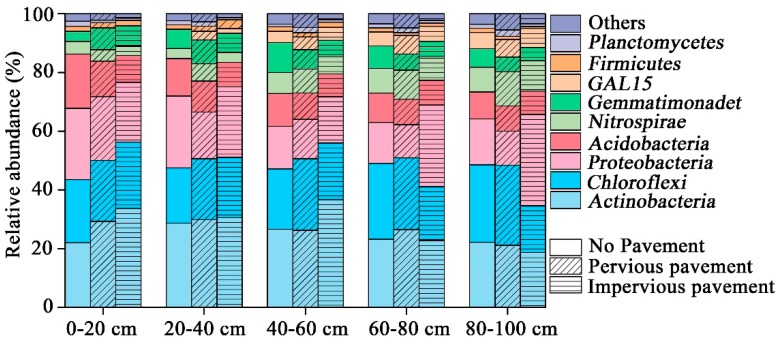
Relative abundance of the dominant bacterial community at the phylum level in samples separated by pavement type category. Relative abundances were found to depend on the average relative number of the bacterial sequences of nine samples from a pavement type. Here, “others” is given with respect to the taxa with a maximum abundance of <0.5% in any sample.

**Figure 2 ijerph-16-01805-f002:**
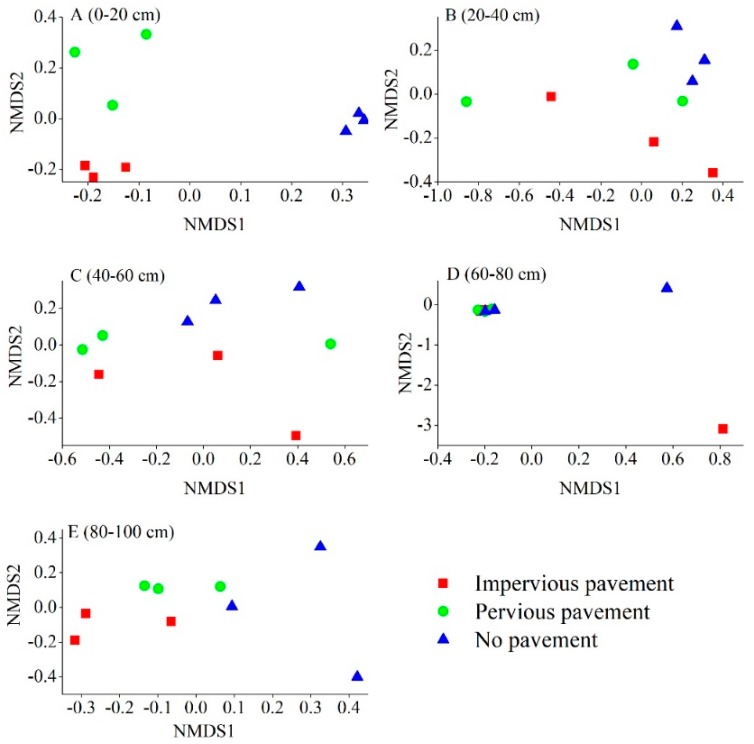
The non-metric multi-dimensional scaling (NMDS) of the bacterial communities comparing all 45 samples from the three pavement types. Sites have been color-coded according to pavement type. (**A**) 0–20 cm; (**B**) 20–40 cm; (**C**) 40–60 cm; (**D**) 60–80 cm; (**E**) 80–100 cm.

**Figure 3 ijerph-16-01805-f003:**
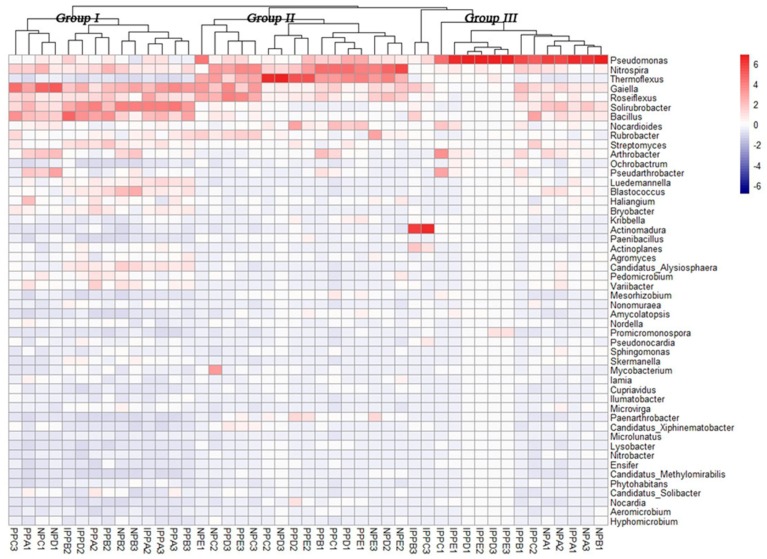
Heatmap of dominant genera of soil bacteria and cluster analysis of bacterial community composition across pavement types. The heatmap and clustering were computed from Operational taxonomic units (OTUs). The blue denotes low relative abundance across a bacterial taxon, and the red denotes high relative abundance.

**Figure 4 ijerph-16-01805-f004:**
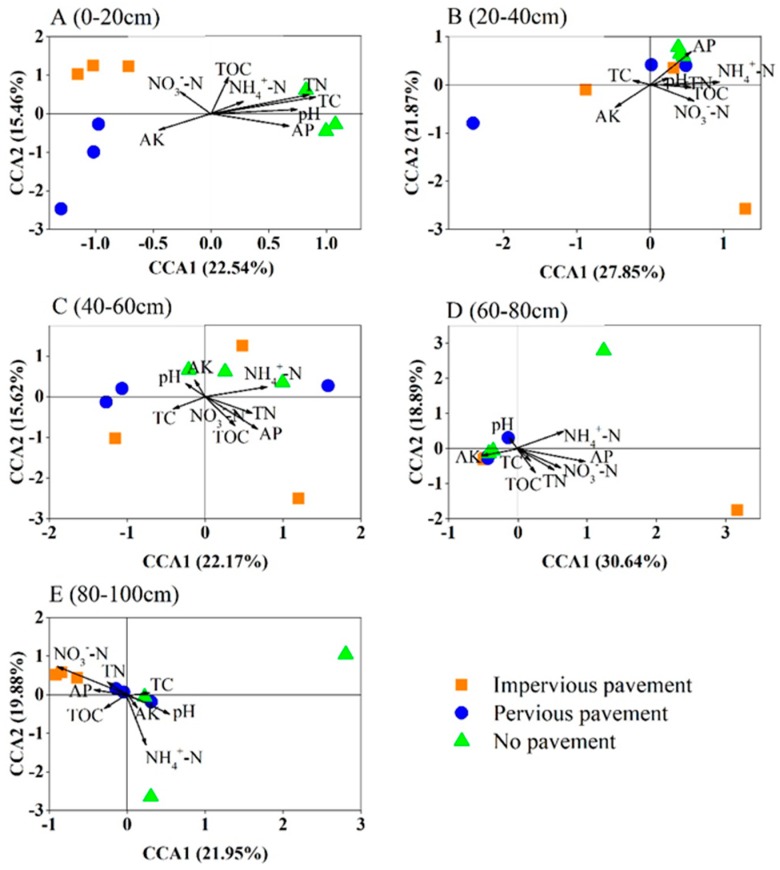
Canonical correspondence analysis (CCA) of the bacterial communities with symbols coded by pavement type under different soil depths. (**A**) 0–20 cm; (**B**) 20–40 cm; (**C**) 40–60 cm; (**D**) 60–80 cm; (**E**) 80–100 cm.

**Table 1 ijerph-16-01805-t001:** The alpha diversity of bacterial communities in different land pavement. Data are expressed as mean ± standard error (SE), *n* = 3. Different letters (a, b and c) indicate statistical significance at *p* < 0.05 across different pavements. PP: pervious pavement; IPP: impervious pavement; NP: non-pavement.

Depth	Pavement	Chao 1 Index	Shannon Index
0–20	IPP	2156.66 ± 321.27 a	10.00 ± 0.07 b
	PP	2109.81 ± 499.54 a	10.14 ± 0.02 b
	NP	2619.07 ± 68.48 a	10.31 ± 0.06 a
20–40	IPP	2225.85 ± 344.99 b	9.53 ± 0.66 a
	PP	2859.2 ± 491.24 a,b	10.03 ± 0.26 a
	NP	3260.76 ± 131.40 a	10.43 ± 0.04 a
40–60	IPP	2300.37 ± 172.86 b	9.55 ± 0.39 a
	PP	2248.56 ± 323.33 b	9.67 ± 0.36 a
	NP	3109.04 ± 74.06 a	10.24 ± 0.08 a
60–80	IPP	1792.47 ± 51.24 b	9.15 ± 0.61 a
	PP	2229.60 ± 308.81 ab	9.75 ± 0.21 a
	NP	2234.90 ± 484.20 a	9.80 ± 0.45 a
80–100	IPP	1668.99 ± 119.23 c	8.87 ± 0.10 b
	PP	2309.48 ± 137.56 b	9.86 ± 0.18 a
	NP	2505.97 ± 310.05 a	10.06 ± 0.15 a

**Table 2 ijerph-16-01805-t002:** Results of stepwise regression for the effects of soil properties on the alpha diversity of bacterial communities. AP, available phosphorus; SOC, soil organic carbon; AK, available potassium.

Index	Regression Model	R^2^	F	*p*
Shannon	y = 10.595 + 0.018 (AP) − 0.101 (SOC) − 0.006 (AK)	0.411	11.252	0.000
Chao 1	y = 3131.595 − 9.262 (NO_3_^−^-N) – 8.432(AK) + 6.863 (AP)	0.273	6.509	0.001

**Table 3 ijerph-16-01805-t003:** Mantel test results for the correlation between community composition and environmental variables for bacteria across pavements. TC, total carbon; TN, total nitrogen. Bold numbers mean ***p*** < 0.05.

Depth (cm)	0–20		20–40		40–60		60–80		80–100	
	R	*p*	R	*p*	R	*p*	R	*p*	R	*p*
pH	0.201	0.095	0.165	0.137	−0.084	0.63	0.185	0.207	−0.177	0.721
TC	**0.407**	**0.024**	−0.111	0.759	0.057	0.359	−0.049	0.546	−0.029	0.551
TN	**0.32**	**0.034**	0.029	0.41	0.283	0.094	0.589	0.051	−0.043	0.493
SOC	0.172	0.126	−0.046	0.531	0.228	0.134	0.458	0.053	0.263	0.123
NH_4_^+^−N	0.049	0.393	0.073	0.298	**0.397**	**0.030**	0.643	0.018	**0.578**	**0.006**
NO_3_^−^−N	−0.064	0.598	0.016	0.421	0.208	0.167	**0.643**	**0.028**	**0.433**	**0.020**
AP	0.2076	0.119	0.148	0.161	**0.502**	**0.003**	**0.781**	**0.002**	−0.263	0.823
AK	0.005	0.443	0.103	0.276	−0.084	0.661	0.184	0.209	−0.177	0.750

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
