# Peer review of "The Effects of Pavement Types on Soil Bacterial Communities across Different Depths"

_ijerph, 2019, doi:10.3390/ijerph16101805_

Round 1

Reviewer 1 Report

Ms is interesting and is very carefully and logically prepared.

Noteworthy are very carefully prepared drawings and results description.

I suggest only minor revision before ms acceptation for publication. My comments are presented below:

Lines 14, 278: should be „microorganisms”

Line 15: the name of Fraxinus Chinensis should be write in italic style

Line 23: abbreviations have to be explained in the place of their first using (also in Abstract)

Line 23: influenced on...

Line 36: should be “evidenced” instead of “show”. Additionally, you wrote :„ Previous studies....” – in this case more than one citation at the end of sentences should be applied

Line 45: please add the following citation here:

Wolińska A., Stępniewska Z., Pytlak A. 2015. The effect of environmental factors on total soil DNA content and dehydrogenase activity. Archives of Biological Sciences, 67 (2), 493-501.

Line 188: should be: the highest...and the lowest...

Line 199: put “demonstrated” instead of “showed”

Line 247: should be NO3-N

Line 255 and 267: please add the following citation:

Wolińska A., Kuźniar A., Zielenkiewicz U., Banach A., Izak D., Stępniewska Z., Błaszczyk M. 2017. Metagenomic analysis of some potential nitrogen-fixing bacteria in arable soils at different formation process. Microbial Ecology, 73: 162-176.

Line 279: should be: Hu et al. [4]...

Line 283: should be: Viswanathan et al. [22]…

Author Response

Dear Reviewer:

 Thank you for your comments concerning our manuscript entitled “The effects of pavement types on soil bacterial communities across different depths” (ID: ijerph-507452). Those comments are all valuable and very helpful for revising and improving our paper, as well as the important guiding significance to our researches. All the comments below have been carefully considered and corresponding revisions have been made in the revised version (see the replies to the following specific comments).

Specific comments

Lines 14, 278: should be microorganisms”

Reply: we have changed “microorganism” into “microorganisms” (line15 and 279).

Line 15: the name of Fraxinus Chinensis should be write in italic style

Reply: “Fraxinus Chinensis” have been written in italic style (line 16-17).

Line 23: abbreviations have to be explained in the place of their first using (also in Abstract

Reply: abbreviations have been explained in the abstract (line 17-24).

Line 23: influenced on...

Reply: we have changed “influenced” into “influenced on” (line 25).

Line 36: should be “evidenced” instead of “show”. Additionally, you wrote :„ Previous studies....” – in this case more than one citation at the end of sentences should be applied

Reply: we have changed “show” into “evidenced” (line 38), and added a citation in line 39.

Line 45: please add the following citation here:

WoliĹ„ska A., StÄ™pniewska Z., Pytlak A. 2015. The effect of environmental factors on total soil DNA content and dehydrogenase activity. Archives of Biological Sciences, 67 (2), 493-501.

Reply: we have cited the citation (line 44).

Line 188: should be: the highest...and the lowest...

Reply: we have revised the sentence as suggested (line 189).

Line 199: put “demonstrated” instead of “showed”

Reply: we have changed “showed” into“demonstrated” (line 200).

Line 247: should be NO3-N

Reply: we have revised the sentence as suggested (line 249).

Line 255 and 267: please add the following citation:

WoliĹ„ska A., KuĹşniar A., Zielenkiewicz U., Banach A., Izak D., StÄ™pniewska Z., BĹ‚aszczyk M. 2017. Metagenomic analysis of some potential nitrogen-fixing bacteria in arable soils at different formation process. Microbial Ecology, 73: 162-176.

Reply: we have cited the citation in line 257 and 269.

Line 279: should be: Hu et al. [4]...

Reply: we have added “[4]” as suggested (line 280).

Line 283: should be: Viswanathan et al. [22]…

Reply: we have added “[22]” as suggested (line 284).

Reviewer 2 Report

1) According to the authors, although it is well known that “… pavements have remarkable effects on topsoil microorganism…”, however, they still claim, in this paper, that “…remains unclear…” “…how subsoil microbial communities respond to pavements…” because  “…the effects of pavement on bacterial communities with soil depth remain  largely unknown…”  and “…our understanding of the structure and diversity of soil microbial communities is limited  primarily to surface horizons, with the vast majority of studies focusing solely on the top 15 cm to 20 cm of the soil column [5,6] ...”.

2) In addition to the previous issues raised by the authors, they also still reclaim that is needed to clarify “…whether pervious pavement can (indeed?) provide better habitat to microbial communities and (as well as to) plants compared to impervious pavement…”???

3) To address these issues, the authors set up, in this paper, a well “purpose-tailored” methodological approach based on a “…high-throughput sequencing of the bacterial 16S rRNA gene …” performed on samples colleted after five years from a “…field experiment…””… across different depths…(as deep as 100 cm below the soil surface)…” and realized in a selected “study site” “…in the Zhangtou village, Changping District, Beijing, China (40°12′N, 116°08′E) …”, to determine both  â€śâ€¦the soil bacterial community composition and  diversity …” as well as which were  “…the main factors that influenced soil bacterial composition and diversity across different pavements…” and depth?

4) Among the several outputs pointed out along their study, the authors have stressed out that “…Understanding the effects of pavements on soil microbial communities across different depth can provide a reference for plant growth in urban areas…”, that seems to be a very interesting point of their research.  

5) The discussion performed by the authors on the data obtained along the paper, has taken into account, on the one hand, several previous works performed by others working on the same field (see for instance, the references cited in the text), as well as, on the other hand, the constrains imposed by the choice they have to make for a “purpose-tailored” methodological approach.  

6) For instance, the authors have stressed out that “their study” provides, to the best of their knowledge, “the FIRST evidence that the effects of pavement on subsoil microbial community are minor in the short-term experiment (i.e., five years of experimentally imposed pavement)”?

7) So taking all these points into account, the authors themselves recognise, however, that “…more experiments are necessary to explore the effects of pavement on deep soils …”?

8) In addition, the authors have finished this paper by claming the need of “…Future studies (that) should investigate how microbial communities respond to pavements over a longer period “as “...these findings will (could??) be highly valuable in  guiding the management of soils and plants in urban areas …”.

9) In spite of the comments that have been made above, minor spell check and wording such as some of the following examples listed below are, however, required:

·        The references to the shorts (abbreviations) for the different pavement types (PP and NP samples, for instance, pg 4, line 157; and see also tables 1 and 2) should be included next to the first references that have been made  to them (see, for instance, the abstract and also the following text);

·        pg 2, line 84, in this statement it seems to have an “extra” “a” that needs to be removed (see for instance, “which a have a similar height”);

·        Table 1 has not been cited in the text as it should be (in its proper place in the text);

·        pg 3,line 97, see also tables 1, 4, and 5, the authors should pay attention to the correct symbol for “the soil pH”;

·        pg 3,line 114, a “ space” should be introduced between “a” and “PicoGreen” words;

·        pg 3, line 155, an “extra” word “were” should de removed in this sentence;

·        pg 3, line 125, the text “Twoway” should be replace by the following “Two-way”;

·        pg 5, line 169, (legend of Figure 2), a “ space” should be introduced between “(NMDS)” and “of the bacterial ….” words;

·        pg 8, Table 1, a legend should be included in order to clarify its content, specially the outputs of the two-way analysis of variance (ANOVA);

·        pg 9, Table 2, a legend might seem to be need for the abbreviations regarding the different pavement types;

·        pgs 10 and 11, Table 5, the authors should pay attention to the abbreviations for the soil properties. They should be the same in all text;

·        pg 12, lines 265 and 266, the wording of this statement should be checked;

Finally, pg 12, line 279,  a reference to “Hu et al.” should be included in the text.

Author Response

Dear Reviewer:

Thank you for your comments concerning our manuscript entitled “The effects of pavement types on soil bacterial communities across different depths” (ID: ijerph-507452). Those comments are all valuable and very helpful for revising and improving our paper, as well as the important guiding significance to our researches. We have studied comments carefully and have made correction which we hope meet with approval. The main corrections in the paper and the responds to your comments are as follows:

1)     According to the authors, although it is well known that “… pavements have remarkable effects on topsoil microorganism…”, however, they still claim, in this paper, that “…remains unclear…” “…how subsoil microbial communities respond to pavements…” because  “…the effects of pavement on bacterial communities with soil depth remain  largely unknown…”  and “…our understanding of the structure and diversity of soil microbial communities is limited primarily to surface horizons, with the vast majority of studies focusing solely on the top 15 cm to20 cm of the soil column [5,6] ...”.

Reply: We have re-written paragraph 1 in introduction according to your suggestion (Line 35-40).

2)  In addition to the previous issues raised by the authors, they also still reclaim that is needed to clarify “…whether pervious pavement can (indeed?) provide better habitat to microbial communities and (as well as to) plants compared to impervious pavement…”???

Reply: According our study, pervious pavements did not provide a better habitat for microorganisms when compared to impervious pavements in the short term (Line 27-30). Because there was no significant difference in bacterial communities between impervious pavement and pervious pavement.

3)  To address these issues, the authors set up, in this paper, a well “purpose-tailored” methodological approach based on a “…high-throughput sequencing of the bacterial 16S rRNA gene …” performed on samples colleted after five years from a “…field experiment…””… across different depths…(as deep as 100 cm below the soil surface)…” and realized in a selected “study site” “…in the Zhangtou village, Changping District, Beijing, China (40°12′N, 116°08′E) …”, to determine both  â€śâ€¦the soil bacterial community composition and  diversity …” as well as which were  “…the main factors that influenced soil bacterial composition and diversity across different pavements…” and depth?

Reply: “across different depths (as deep as 100 cm below the soil surface)” was added (Line 67-69).

4)  Among the several outputs pointed out along their study, the authors have stressed out that “…Understanding the effects of pavements on soil microbial communities across different depth can provide a reference for plant growth in urban areas…”, that seems to be a very interesting point of their research.  

Reply: Yes. Because soil microorganisms are benefit to plants uptalking nutrients. Our results indicated that pavements changed soil bacterial communities at topsoil layer, which can provide a reference for plant growth.

5)  The discussion performed by the authors on the data obtained along the paper, has taken into account, on the one hand, several previous works performed by others working on the same field (see for instance, the references cited in the text), as well as, on the other hand, the constrains imposed by the choice they have to make for a “purpose-tailored” methodological approach.  

Reply: Our study designed only one type of pervious pavements, and got an opposite conclusion with previous study. More types of pervious pavement should be study for verifying our conclusions.

6)  For instance, the authors have stressed out that “their study” provides, to the best of their knowledge, “the FIRST evidence that the effects of pavement on subsoil microbial community are minor in the short-term experiment (i.e., five years of experimentally imposed pavement)”?

Reply: “first” was deleted (Line 241-242).

7)  So taking all these points into account, the authors themselves recognise, however, that “…more experiments are necessary to explore the effects of pavement on deep soils …”?

Reply: Yes, we think it is necessary to explore the response of bacterial community to pavement over long time. Because our results differed with some studies that studied the effects of pavement on soil microorganisms after disturbed by pavement for more than 10 years.

8)  In addition, the authors have finished this paper by claming the need of “…Future studies (that) should investigate how microbial communities respond to pavements over a longer period “as “...these findings will (could??) be highly valuable in guiding the management of soils and plants in urban areas …”.

Reply: Our study indicated that pavements had a greater impact on topsoil bacterial communities. Soil microorganisms had close relationship with roots. Our results can provide reference to manage soils as well as plants in urban areas.

9) In spite of the comments that have been made above, minor spell check and wording such as some of the following examples listed below are, however, required:

 The references to the shorts (abbreviations) for the different pavement types (PP and NP samples, for instance, pg 4, line 157; and see also tables 1 and 2) should be included next to the first references that have been made  to them (see, for instance, the abstract and also the following text);

Reply: we have added abbreviations for the different pavement types as suggested (Line 157-158) 

·pg 2, line 84, in this statement it seems to have an “extra” “a” that needs to be removed (see for instance, “which a have a similar height”); 

Reply: we have deleted "a" (Line 83).

Table 1 has not been cited in the text as it should be (in its proper place in the text);

Reply: we cited Table 1 in the line 202.

 pg 3,line 97, see also tables 1, 4, and 5, the authors should pay attention to the correct symbol for “the soil pH”;

Reply: we have changed “PH” into “pH” in table 1 (line 221), 4 (line 230), and 5 (line 233).

 pg 3,line 114, a â€ś space” should be introduced between “a” and “PicoGreen” words;

Reply: we added a “space” between“a” and “PicoGreen” (line 113).

 pg 3, line 155, an “extra” word “were” should de removed in this sentence;

Reply: we have deleted " were " in line 156.

 pg 3, line 125, the text “Twoway” should be replace by the following “Two-way”;

Reply: we have changed “Twoway” into “Two-way” in table 1 (line 126).

pg 5, line 169, (legend of Figure 2), a “ space” should be introduced between “(NMDS)” and “of the bacterial ….” words;

Reply: we added a “space” between“(NMDS)” and “of the bacterial” (line 170).

 pg 8, Table 1, a legend should be included in order to clarify its content, specially the outputs of the two-way analysis of variance (ANOVA)

Reply: we added a legend regarding the different pavement types in Table 1 (line 221).

pg 9, Table 2, a legend might seem to be need for the abbreviations regarding the different pavement types;

Reply: we added a legend regarding the different pavement types in Table 2 (line 224).

 pgs 10 and 11, Table 5, the authors should pay attention to the abbreviations for the soil properties. They should be the same in all text;

Reply: we have revised the abbreviations of soil properties in Table 5 (line 233-234).

pg 12, lines 265 and 266, the wording of this statement should be checked;

Reply: we have revised the sentence (line 267-268).

Finally, pg 12, line 279,  a reference to “Hu et al.” should be included in the text.

Reply: we have revised the sentence as suggested (line 279).